# Pruning Feature Extractor Stacking for Cross-domain Few-shot Learning

**Hongyu Wang**                                                    *hongyu.wang@waikato.ac.nz*
*Department of Computer Science*
*University of Waikato*

**Eibe Frank**                                                     *eibe.frank@waikato.ac.nz*
*Department of Computer Science*
*University of Waikato*

**Bernhard Pfahringer**                                            *bernhard.pfahringer@waikato.ac.nz*
*Department of Computer Science*
*University of Waikato*

**Geoffrey Holmes**                                                *geoffrey.holmes@waikato.ac.nz*
*Department of Computer Science*
*University of Waikato*

**Reviewed on OpenReview:** *https://openreview.net/forum?id=p499xXaclC*

## Abstract

Combining knowledge from source domains to learn efficiently from a few labelled instances in a target domain is a transfer learning problem known as cross-domain few-shot learning (CDFSL). Feature extractor stacking (FES) is a state-of-the-art CDFSL method that maintains a collection of source domain feature extractors instead of a single universal extractor. FES uses stacked generalisation to build an ensemble from extractor snapshots saved during target domain fine-tuning. It outperforms several contemporary universal model-based CDFSL methods in the Meta-Dataset benchmark. However, it incurs higher storage cost because it saves a snapshot for every fine-tuning iteration for every extractor. In this work, we propose a bidirectional snapshot selection strategy for FES, leveraging its cross-validation process and the ordered nature of its snapshots, and demonstrate that a 95% snapshot reduction can be achieved while retaining the same level of accuracy.

## 1 Introduction

Deep learning methods generally require a large amount of labelled training data to obtain high predictive performance when training a model for a new target domain. Cross-domain few-shot learning (CDFSL) attempts to address this by transferring knowledge gained from one or more source domains, with plenty of labelled training instances, to the target domain, for which only few labelled training instances are available. The target domain may only be weakly related to any particular source domain. CDFSL aims to achieve high accuracy when predicting labels for new unlabelled instances in the target domain—the so-called "query set"—by aggregating knowledge from the source domains and placing emphasis on those most relevant to the small labelled target domain training set—the so-called "support set". In practice, to evaluate CDFSL methods and compare them to each other, a large collection of query-and-support set pairs is sampled from a large labelled dataset such as CIFAR10, and average predictive performance on the query sets is reported. Each query-and-support set pair is called an "episode".

Most recent top-performing CDFSL algorithms, such as URL (Li et al., 2021), its TSA-based variant (Li et al., 2022), and FLUTE (Triantafillou et al., 2021), derive a universal model from source domains and

then fit it to a support set using a small set of trainable parameters. In contrast, FES (Wang et al., 2024), a CDFSL method based on stacked generalisation (Wolpert, 1992), maintains a collection of source domain-specific models instead of a universal one and produces a linear combination of model snapshots saved during support set fine-tuning that have been evaluated through cross-validation. FES outperforms several contemporary universal model-based methods on the Meta-Dataset benchmark (Triantafillou et al., 2020), achieving the current state of the art (Wang et al., 2024). FES can be applied with semi-supervised STC classifiers (Wang et al., 2023) to further improve its accuracy using additional unlabelled data. FES also exhibits good interpretability as each snapshot's weight reflects its contribution to the learned task. However, it has high storage cost as it saves a snapshot for every fine-tuning iteration for every model. In this paper, we address this shortcoming by taking inspiration from the results in Wang et al. (2024), which show that only a minority of snapshots are assigned significant weights: most are assigned trivial weights and have no meaningful impact on predictions individually.

The key observation for the work presented here is that snapshot pruning in FES can be framed as a feature subset selection problem where the logits provided by every snapshot are the input features for the linear combination trained in the stacking approach—the "stacking classifier". Wrapper-based approaches using greedy search methods such as forward selection or backward elimination can be applied to such problems by using cross-validation performance to determine the relevance of feature subsets (Kohavi & John, 1997). However, naive greedy search yields an infeasibly large search space when applying FES with a non-trivial number of snapshots, as it evaluates combinations of a selected subset with each of the remaining features, yielding runtime that scales quadratically in the total number of snapshots. Fortunately, we can exploit the ordering of snapshots, provided by the sequence of fine-tuning steps for each extractor, to limit the set of available choices, only evaluating a small pool of potentially relevant candidates in each step of the search. This strategy can be interpreted as a form of linear forward selection (Gütlein et al., 2009).

We propose a bidirectional snapshot selection (BSS) strategy for pruning FES that performs linear forward selection with an initial candidate pool comprising snapshots at the beginning and the end of every model's fine-tuning. When a candidate is selected, the pool is replenished with the candidate's adjacent snapshot in fine-tuning. Given a fixed number of extractors, BSS runtime scales linearly in the total number of snapshots. We show BSS outperforms several baseline strategies and achieves over 95% average snapshot reduction while retaining the same level of accuracy as unpruned FES in both supervised and semi-supervised settings.

## 2 Related Work

We first review work in CDFSL before covering FES. Lastly, we consider wrapper-based feature selection. To keep this review concise, we do not discuss other, potentially orthogonal, machine learning approaches for settings where it is expensive to label data, such as semi-supervised learning, exemplified by the classic self-training approach (Chapelle et al., 2006), or active learning, exemplified by uncertainty sampling (Lewis & Gale, 1994) and query by committee (Seung et al., 1992).

### 2.1 Cross-domain few-shot learning

Early few-shot learning work is in-domain (Vinyals et al., 2016; Snell et al., 2017; Finn et al., 2017), where source and target domain classes are different partitions of the same dataset. Guo et al. (2020) showed that simple transfer learning outperforms these methods in CDFSL. Meta-Dataset (Triantafillou et al., 2020) is a benchmark for evaluating CDFSL methods, originally containing eight domains, ilsvrc_2012, omniglot, aircraft, cu_birds, dtd, quickdraw, fungi, and vgg_flower, and two target domains from which episodes can be sampled: traffic_sign and mscoco. Three additional target domains—mnist, cifar10, and cifar100—were added by Requeima et al. (2019), and a further five, namely, CropDisease, EuroSAT, ISIC, ChestX, and Food101, were added by Wang et al. (2024). Note that only target domain tasks qualify as cross-domain. Using established terminology, an algorithm's accuracy in the target domains represents its "strong generalisation" (SG) performance, while that in source domains, even when measured on episodes sampled from data of these domains that has been held out, represents performance in "weak generalisation" (WG).

CDFSL can be divided into three stages: pretraining, fine-tuning, and inference. The pretraining stage fits model(s) to source domain data. This stage generally has abundant training data in one or multiple source domains. Some CDFSL methods pretrain a model specifically on each source domain, while others pretrain one model on all source domains. The fine-tuning stage fits the pretrained model(s) to few-shot target domain data. The inference stage uses the fine-tuned model(s) for predictions in the target domain.

Early work in CDFSL includes SUR (Dvornik et al., 2020), URT (Liu et al., 2021), CNAPs (Requeima et al., 2019), simple CNAPs (Bateni et al., 2020), and transductive CNAPs (Bateni et al., 2022). URL (Li et al., 2021) and FLUTE (Triantafillou et al., 2021) are newer methods that outperform these algorithms on Meta-Dataset. URL distils a universal model by optimising it to match feature and logit output of source domain extractors using source domain instances. Given an episode, a feature projection attached at the end of the universal model is fine-tuned on the support set with a nearest-centroid classifier. In contrast, FLUTE trains a universal template with source domain-specific batch normalisation layers and a set of convolutional layers shared by all source domains. Given an episode, a set encoder (Zaheer et al., 2017) blends the batch normalisation weights based on the predicted likeliness of the support set to the source domains. The blended weights are fine-tuned on the support set.

Task-specific adaptors (TSA) (Li et al., 2022), originally proposed for URL, are architectural components specifically designed to improve fine-tuning for CDFSL. Given a feature extractor, these adaptors are attached to its convolutional layers. Multiple adaptor configurations are viable, and Li et al. (2022) found residual channel-wise projections to be most effective, fine-tuned along with the feature projection from URL. We use TSA with URL as the few-shot reference method because it outperforms the original URL and FLUTE (Li et al., 2022). We also use TSA for obtaining the feature extractor snapshots in FES, as in Wang et al. (2024).

## 2.2 Feature extractor stacking

FES (Wang et al., 2024) composes an ensemble of source domain-specific feature extractor snapshots fine-tuned on the support set, using stacking to estimate each snapshot's performance via cross-validation and combine all snapshots linearly. To obtain snapshots in FES for a given episode, each feature extractor is combined with a classifier and fine-tuned on the support set, and a snapshot of the extractor and classifier is saved before any fine-tuning and after each fine-tuning iteration. To yield a linear combination of the snapshots by training the stacking classifier, cross-validation is used: the support set is split into two partitions using stratified cross-validation, and these two partitions alternate as training and test splits to produce logits, where an extractor and its classifier are fine-tuned on one split and produce logits from the other split after each fine-tuning iteration. These logits are used as training data for the weights of the stacking classifier, constituting a linear combination of snapshots. The trained stacking classifier is then applied to combine the logits of the snapshots obtained by fine-tuning on the *full* support set to obtain classifications for the query set. The weights of the stacking classifier can be visualised as a heatmap to indicate each snapshot's impact on final predictions. In essence, FES is a stacked generalisation (Wolpert, 1992) method adopting stratified two-fold cross-validation and treating each snapshot as a separate base model.

Wang et al. (2023) showed that self-trained centroids (STC) can be applied to adapt FES for semi-supervised learning and improve its accuracy by utilising unlabelled instances. Here, a semi-supervised CDFSL episode comprises a labelled support set and an unlabelled set pertaining to the same classes. The supervised fine-tuning procedure is first performed using the labelled set to obtain fine-tuned feature extractor(s) with which the labelled and unlabelled sets are then converted into feature vectors. Self-training (Rosenberg et al., 2005) is applied to the centroid-based classification head based on the extracted feature vectors: 1) the centroids of labelled feature vectors are used to soft-label the unlabelled feature vectors with pseudo labels, 2) the unlabelled feature vectors are aggregated to obtain class centroids by forming a weighted average using their soft pseudo labels, and 3) the two sets of class centroids from the labelled and unlabelled feature vectors are combined using a simple arithmetic average, leading to semi-supervised self-trained centroids for classifying query instances. STC can be considered a generalisation of the "inference only" baseline in Ren et al. (2018).

For FES, STC is applied to each extractor during both cross-validation and full support set fine-tuning. During cross-validation, STC is applied using the training split as the labelled set, and test split logits produced by the self-trained centroid classifier are used as training data for the FES stacking classifier. In

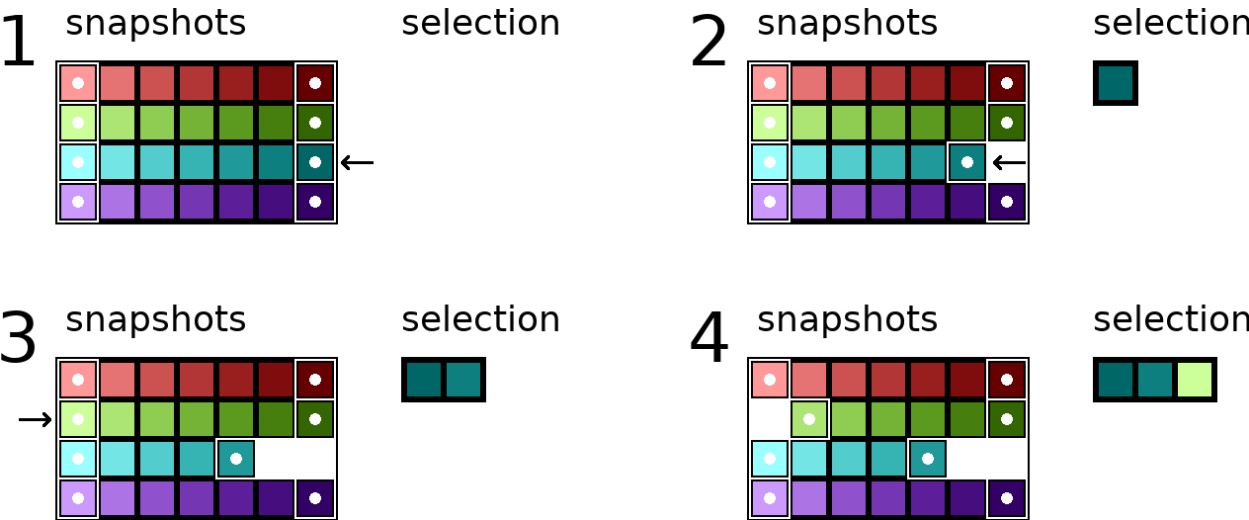

Figure 1: Bidirectional snapshot selection: The candidate pool is marked with light circles. The arrow marks the selected snapshot yielding the highest decrease in cross-validated loss. Its neighbour replaces it in the pool.

short, compared to supervised FES, semi-supervised FES with STC uses logits obtained from STC classifiers instead of supervised centroid classifiers whenever logits are needed.

### 2.3 Wrapper-based feature selection

Greedy feature subset selection using wrapper methods that apply cross-validation was popularised by seminal work presented in Kohavi & John (1997). Commonly, it is implemented using forward selection as the search strategy, so it starts with an empty subset of features. Then, given a set of candidate features, it separately includes each candidate in the existing subset and evaluates the subset's features in concert using cross-validation. The candidate whose inclusion leads to the highest cross-validation performance is selected into the feature subset. Intuitively, the feature with the highest cross-validation performance on its own is first selected into the empty subset, then the one with the highest cross-validation performance in concert with the first selection is selected into the subset, and so on. Commonly, the search proceeds until no candidate improves cross-validation performance when added to the existing feature subset, and this subset is returned as the resulting feature selection. This approach of evaluating a candidate jointly with the existing subset is more robust against correlated features than evaluating the candidate independently, as a feature needs to yield the highest improvement of the existing subset's predictive performance to merit its selection. Note that consecutive FES snapshots in fine-tuning are inherently correlated.

Linear forward selection is a variant of forward selection that only evaluates a limited pool of candidates at each step (Gütlein et al., 2009). Given a feature set without explicit order, this method ranks the features by their individual cross-validation performance and initialises a pool of a user-specified size with the top-ranked features. During the search, this pool can be optionally replenished by the remaining features in ranked order. Gütlein et al. (2009) found linear forward selection is faster, finds smaller subsets, and sometimes improves accuracy compared to naive forward selection. As FES snapshots of the same extractor are implicitly ordered because they are sequentially generated during fine-tuning, this search technique is a natural fit for computationally efficient pruning of FES ensembles.

## 3 Pruning FES with BSS

We now present bidirectional snapshot selection (BSS) for pruning FES ensembles, which is shown as pseudo code in Algorithm 1 and visualised in Figure 1. In essence, BSS initialises an empty selection and a pool of candidates containing the first and last fine-tuning snapshots of each extractor; it iteratively selects the candidate snapshot from the pool into the selection that leads to the highest reduction in cross-validation

---

**Algorithm 1** Bidirectional snapshot selection

---

**Input**: Support set instances $X$ and labels $Y$, $K$ extractors $\Phi_1, ..., \Phi_K$
**Parameter**: fine-tuning iterations $J$ and patience $P$
**Output**: Snapshot subset $\mathbb{S}$

1: Split the support set into $X_1, X_2$ and $Y_1, Y_2$ using stratified cross-validation.
2: **for** $k \in [1, K]$ **do**
3:     Fine-tune $\Phi_k$ using $X_1$ and $Y_1$ and save snapshots $\Phi_{k,0}^{CV_1}, ..., \Phi_{k,J}^{CV_1}$.
4:     Fine-tune $\Phi_k$ using $X_2$ and $Y_2$ and save snapshots $\Phi_{k,0}^{CV_2}, ..., \Phi_{k,J}^{CV_2}$.
5: **end for**
6: Let selected subset $\mathbb{S} = \varnothing$, temporary subset $\mathbb{T} = \varnothing$, best loss $\ell_{best} = \infty$, patience value $PV = P$, candidate pool $\mathbb{P} = \{(k,j)|k \in [1, K], j \in \{0, J\}\}$.
7: **while** $\mathbb{P} \neq \varnothing \wedge PV \geq 0$ **do**
8:     Let temporary best cross-validation loss $\ell_{temp} = \infty$.
9:     **for** $p \in \mathbb{P}$ **do**
10:         Apply $\{\Phi_e^{CV_1}|e \in \mathbb{T} \cup \{p\}\}$ to $X_2$ to extract logits $L^{CV_2}$.
11:         Apply $\{\Phi_e^{CV_2}|e \in \mathbb{T} \cup \{p\}\}$ to $X_1$ to extract logits $L^{CV_1}$.
12:         Train FES stacking classifier weights $W^{CV_1}$ using $L^{CV_1}$ and $Y_1$ and evaluate it using $L^{CV_2}$ and $Y_2$ to obtain loss $\ell_{CV_2}$.
13:         Train FES stacking classifier weights $W^{CV_2}$ using $L^{CV_2}$ and $Y_2$ and evaluate it using $L^{CV_1}$ and $Y_1$ to obtain loss $\ell_{CV_1}$.
14:         **if** $\ell_{CV_1} + \ell_{CV_2} < \ell_{temp}$ **then**
15:             Let best candidate $b = p$, $\ell_{temp} = \ell_{CV_1} + \ell_{CV_2}$.
16:         **end if**
17:     **end for**
18:     Let $\mathbb{T} = \mathbb{T} \cup \{b\}$; $\mathbb{P} = (\mathbb{P} \cup Pre_b \cup Suc_b) \setminus \mathbb{T}$.
19:     **if** $\ell_{temp} < \ell_{best}$ **then**
20:         Let $\mathbb{S} = \mathbb{T}$, $\ell_{best} = \ell_{temp}$, $PV = P$.
21:     **else**
22:         Let $PV = PV - 1$.
23:     **end if**
24: **end while**
25: **return** $\mathbb{S}$

---

loss when incorporated into the FES ensemble, replenishing the pool with adjacent unexplored snapshots; the iterative process stops when no snapshot in the pool reduces cross-validation loss or no further snapshots remain, and the selection is returned as the final ensemble.

Given a support set containing instances $X$ and labels $Y$, $K$ source domain feature extractors $\Phi_1, ..., \Phi_K$, each to be fine-tuned for $J$ iterations, and a patience value $P \in \mathbb{N}$, BSS returns a subset of the $K \times (J + 1)$ snapshots.

BSS leverages the 2-fold stratified cross-validation performed by FES to guide the search, where the support set is split into two partitions of instances, which alternate to serve for extractor fine-tuning and logit extraction, leading to two partitions of logits. Fine-tuning extractors is generally the most computationally heavy process of FES, and BSS can utilise its outcome without extra cost. BSS alternates the two logit partitions to use one to train the FES stacking classifier weights and the other as input to the trained stacking classifier to obtain meta-logits. The meta-logits are used with their corresponding labels to compute cross-validated cross-entropy, and losses of the two partitions are summed to represent cross-validation loss of the evaluated subset. Selection of subsets is based on minimising this cross-validation loss.

The search starts with an empty subset and initialises its candidate pool with the first and last snapshots of each extractor's fine-tuning process, i.e., $2 \times K$ snapshots. Then, it evaluates each candidate by measuring cross-validation loss on an expanded version of the existing subset (initially empty) to which the candidate has been added. The candidate that leads to the highest decrease in cross-validation loss is selected into the subset and removed from the pool; an adjacent snapshot is used to replenish the pool. This snapshot is the selected candidate's successor if it is at the beginning of fine-tuning or predecessor if it is at the end, and it is only added to the pool if not already in the subset. For brevity, in Algorithm 1, given a snapshot of the $k$-th extractor at the $j$-th iteration $b = (k, j)$, its predecessor set is denoted as $Pre_b = \{(k, j - 1)\}$ if $j - 1 \geq 0$ else $\varnothing$, and its successor set is denoted as $Suc_b = \{(k, j + 1)\}$ if $j + 1 \leq J$ else $\varnothing$.

If the patience parameter $P$ has value zero, the search terminates, and the current subset is returned if no candidate in the pool decreases cross-validation loss when added to the subset. For $P > 0$, the process only terminates if no loss decrease is achieved in $P$ consecutive steps. In this case, the candidate leading to the lowest loss increase is tentatively selected even if a previous subset achieved a lower loss, and its selection is only confirmed when a later step within the patience limit achieves lower loss than the previous best subset.

Termination returns the subset with the lowest cross-validation loss explored by the search. Depletion of the candidate pool also leads to termination of search.

Once BSS returns a subset, the final FES model is trained on this subset only: weights for the stacking classifier are trained on cross-validation logits of the snapshots in the subset; when producing predictions, these weights are used to aggregate the output of the corresponding subset of snapshots fine-tuned using all support set data.

BSS is proposed to address the complexity issue of naïvely selecting from all snapshots, which can be prohibitive for long snapshot sequences. At the beginning, selecting from all snapshots has to perform $J$ times as many evaluations as BSS to select each snapshot. On average, the complexity for the number of evaluations is $O(KJ)$ for selecting each snapshot from all snapshots, while it is $O(K)$ for selecting each snapshot using BSS.

## 4  Experimental Setup

We perform evaluation on Meta-Dataset with the extended set of target domains and adhere to the official sampling method to generate 600 few-shot episodes for each domain: each episode contains 5 to 50 classes, up to 100 support set instances per class, up to 500 (potentially class-imbalanced) support set instances in total, and 10 query set instances per class. The sampled episodes are cached to ensure that all pruning strategies are evaluated using the same episodes.

We use FES with TSA fine-tuning to evaluate pruning strategies. FES hyperparameters are consistent with Wang et al. (2024), and TSA hyperparameters are consistent with Li et al. (2022). ResNet18 (He et al., 2016) extractors are used throughout. Each of the eight source domain extractors is fine-tuned with TSA for 40 iterations, leading to $8 \times 41 = 328$ snapshots in total. We also evaluate semi-supervised FES with TSA fine-tuning and STC, using the hyperparameters from Wang et al. (2023).

We evaluate three baselines: 1) FES without pruning, 2) exhaustively searching through all *extractor* combinations, where selecting an extractor leads to inclusion of all of its 41 snapshots, and 3) exhaustively searching through all combinations of the extractors' *final* snapshots. For reference, we include a URL model with TSA fine-tuning (Li et al., 2022). We evaluate BSS with patience = 0 as our main method. In an ablation study, we evaluate two unidirectional snapshot selection strategies that are akin to BSS but only search in one direction: unidirectional forward snapshot selection (UFSS) initialises its candidate pool with only snapshots from the beginning of fine-tuning, and conversely, unidirectional backward snapshot selection (UBSS) initialises its candidate pool with only snapshots from the end of fine-tuning. We also evaluate BSS, UFSS, and UBSS with patience values from 0, i.e., no patience, to 328, which ensures depletion of the candidate pool. We perform paired $t$-tests with $p = 0.05$ using accuracy of individual episodes to determine statistical significance of differences in accuracy on individual datasets.

## 5  Results

We first compare BSS to the baseline strategies. Then, we visualise pruning results as heatmaps. Following this, we conduct an ablation study with UFSS and UBSS and consider varying patience values. We then evaluate BSS pruning in semi-supervised learning with STC. Lastly, we analyse runtime and memory consumption of FES pruned with BSS and discuss the trade-off between accuracy and cost.

### 5.1  Comparison to baselines

Table 1 compares BSS to the baselines. Accuracy and percentage of snapshots remaining after pruning, i.e., $|\mathbb{S}|/(K \times (J+1))$, are reported with 95% confidence intervals. Mean WG and SG performance and ranks are also reported. Each baseline is evaluated against BSS in paired $t$-tests. If a statistical significance is detected, i.e., $p < 0.05$, ● indicates better BSS accuracy, whereas ○ indicates better baseline accuracy.

Compared to naive FES without pruning, BSS achieves higher average SG accuracy while the two exhaustive baselines perform worse. Exhaustive extractor search only removes approximately half of all snapshots. On

the other hand, while exhaustively searching through combinations of the last snapshots achieves impressive reduction, as only 8 snapshots out of the total 328 are considered, its SG performance is the weakest among the four methods. BSS achieves the highest accuracy in SG while achieving over 95% snapshot reduction. The URL model with TSA fine-tuning achieves the highest WG accuracy but the lowest SG accuracy compared to the pruned and unpruned FES methods.

We have also evaluated several other baselines but they proved inferior to the strategies presented in Table 1, so they are omitted. These baseline strategies include: 1) exhaustive search involving only each extractor's snapshot with the lowest individual cross-validation loss, 2) greedy forward selection with at most one snapshot per extractor, performed by including all snapshots in the initial pool and when a snapshot is selected, removing all candidates of the same extractor, 3) greedy forward selection with at most one snapshot per extractor but allowing replacement of a snapshot in the selected subset with another snapshot of the same extractor, 4) ranking all snapshots by individual cross-validation loss, iterating through the ranking, and selecting snapshots whose inclusion in the subset decreases cross-validation loss, and 5) pruning by applying L1 regularisation to FES stacking classifier training to force the weights of some snapshots to zero, hereby removing them.

## 5.2   Visualisation

Figure 2 presents stacking classifier weight matrices as heatmaps. In each pair of graphs, the left one represents average weight assigned to each snapshot by unpruned FES, and the right one represents the version by FES pruned with BSS. (Unselected snapshots receive zero weight.) Results are averages from the 600 episodes of the corresponding target domain. Five domains are presented in Figure 2 and analysed in detail: traffic_sign, mscoco, ISIC, ChestX, and Food101. The other five target domains are omitted as their unpruned weight matrices focus on one or two source domains where weights change gradually as fine-tuning progresses, while the pruned version focuses on the same domains with sparse weights concentrating at the end of fine-tuning.

In traffic_sign, three source domains are focused on: ImageNet, textures (dtd), and quickdraw. The unpruned version features gradual snapshot weight changes in these three source domains, while the pruned version heavily focuses on the snapshots at the end of fine-tuning, which is sufficient according to Table 1: the pruned version achieves significantly higher accuracy according to a paired $t$-test.

ImageNet is the main focus in both mscoco and Food101. Unpruned FES focuses heavily on early snapshots in mscoco, while pruned FES focuses on the final snapshot. In Food101, the unpruned version focuses on two snapshots in particular: the one before any fine-tuning and the one after two fine-tuning iterations, whereas the pruned version focuses on the former only. Table 1 shows that the pruned version performs significantly better in these two domains than the unpruned version. Pruning also makes FES heatmaps more concise and thus more interpretable in these two domains as well as the aforementioned traffic_sign.

FES appears unfocused in ISIC and ChestX, likely because there is no strong relationship between any source domain and these two target domains. Pruned FES has weights concentrated at the beginning and end of fine-tuning, and Table 1 shows that BSS pruning significantly decreases average accuracy in these two domains. The unpruned heatmaps show significant weights assigned to snapshots in the middle of fine-tuning, which may be the reason for this. The results indicate that the existence of target domains that are related to the source domains can lead to a smoother and more monotonic cross-validation accuracy curve during fine-tuning, resulting in a concentration of the best snapshots at the beginning or the end of fine-tuning, which BSS can leverage to achieve significant pruning without tangible accuracy loss. When such properties are potentially absent, like in ISIC or ChestX, BSS is more likely to suffer a loss in accuracy.

Overall, these figures show that BSS is able to remove many snapshots while retaining and sometimes even enhancing interpretability. Moreover, snapshots at the beginning and the end of fine-tuning can both be relevant to a task. However, it may occasionally fail to include relevant snapshots observed during the middle of fine-tuning.

Table 1: BSS compared to the baselines: no pruning (full), exhaustively searching through extractors (EE), and exhaustively searching through the last snapshot of each extractor (EL). A URL extractor with TSA fine-tuning is evaluated for reference.

| Dataset | BSS accuracy | BSS ratio | full accuracy | EE accuracy | EE ratio | EL accuracy | EL ratio | URL-TSA accuracy |
|---|---|---|---|---|---|---|---|---|
| ilsvrc_2012 | 56.35±1.16 | 3.76±0.28 | 56.22±1.14 | 56.28±1.15 | 50.44±1.34 | 56.32±1.17 | 1.44±0.03 | **56.78±1.12** ○ |
| omniglot | 93.09±0.70 | 12.29±0.81 | 93.33±0.64 ○ | 93.25±0.66 | 47.50±1.86 | **93.33±0.66** ○ | 1.28±0.04 | **94.98±0.41** ○ |
| aircraft | 87.75±0.79 | 4.86±0.36 | 87.64±0.79 | 87.60±0.83 | 50.15±1.31 | 87.72±0.83 | 1.32±0.03 | **88.43±0.51** ○ |
| cu_birds | 80.05±0.86 | 4.03±0.26 | 79.93±0.85 | 80.02±0.85 | 48.88±1.28 | 79.77±0.87 ● | 1.36±0.03 | **81.47±0.72** ○ |
| dtd | 76.48±0.82 | 3.23±0.26 | 76.21±0.81 ● | 76.17±0.80 ● | 49.67±1.27 | 76.69±0.82 ○ | 1.29±0.03 | **77.11±0.67** ○ |
| quickdraw | **83.56±0.59** | 2.93±0.18 | 83.38±0.61 ● | 83.35±0.61 ● | 42.04±1.28 | 83.33±0.62 ● | 1.27±0.03 | 81.96±0.61 ● |
| fungi | 69.50±1.14 | 3.24±0.26 | 69.43±1.13 | **69.67±1.13** ○ | 43.06±1.39 | 69.24±1.13 ● | 1.33±0.03 | 68.30±1.06 ● |
| vgg_flower | 91.85±0.70 | 5.62±0.49 | 91.92±0.66 | 91.88±0.67 | 53.38±1.30 | 92.06±0.67 ○ | 1.45±0.03 | **92.07±0.55** |
| WG avg | 79.83 | 4.99 | 79.74 | 79.78 | 48.10 | 79.80 | 1.34 | **80.14** |
| WG rank | 2.97 | | 2.98 | **2.92** | | 3.02 | | 3.11 |
| traffic_sign | 85.66±0.95 | 2.85±0.31 | 84.90±0.97 ● | 84.80±0.98 ● | 59.56±1.52 | **85.72±0.94** ○ | 1.45±0.03 | 82.81±0.94 ● |
| mscoco | **54.54±1.03** | 3.38±0.22 | 54.14±1.02 ● | 54.28±1.03 ● | 59.50±1.59 | 53.63±1.04 ● | 1.51±0.03 | 53.81±1.06 ● |
| mnist | 96.99±0.52 | 6.35±0.63 | **97.08±0.48** ○ | 96.85±0.52 ● | 40.73±1.20 | 97.01±0.52 | 1.19±0.03 | 96.62±0.38 ● |
| cifar10 | 78.32±0.89 | 2.65±0.21 | 78.14±0.87 | 78.24±0.87 | 46.33±1.49 | 78.52±0.86 ○ | 1.24±0.03 | **79.94±0.76** ○ |
| cifar100 | **70.64±1.06** | 3.55±0.33 | 70.35±1.05 ● | 70.32±1.05 ● | 49.40±1.54 | 70.62±1.06 | 1.49±0.03 | 70.26±1.03 ● |
| CropDisease | 87.96±0.68 | 4.57±0.35 | **88.15±0.68** ○ | 87.94±0.68 | 56.83±1.20 | 87.83±0.68 ● | 1.58±0.03 | 84.38±0.77 ● |
| EuroSAT | 89.33±0.64 | 2.50±0.20 | 88.85±0.63 ● | 88.97±0.61 ● | 41.33±1.33 | 89.37±0.62 | 1.21±0.03 | **89.62±0.54** |
| ISIC | 48.32±0.94 | 3.42±0.17 | **49.52±0.93** ○ | 49.16±0.95 ○ | 56.33±1.38 | 47.82±0.95 ● | 1.47±0.03 | 48.38±0.88 |
| ChestX | 27.04±0.58 | 3.20±0.33 | **27.67±0.61** ○ | 27.55±0.60 ○ | 55.35±1.60 | 26.69±0.59 ● | 1.37±0.03 | 27.17±0.57 |
| Food101 | **55.71±1.12** | 2.94±0.19 | 55.22±1.11 ● | 55.35±1.11 ● | 48.71±1.16 | 54.58±1.15 ● | 1.61±0.03 | 53.37±1.21 ● |
| SG avg | **69.44** | 3.55 | 69.40 | 69.36 | 51.43 | 69.17 | 1.41 | 68.64 |
| SG rank | **2.76** | | 2.90 | 2.94 | | 3.00 | | 3.40 |

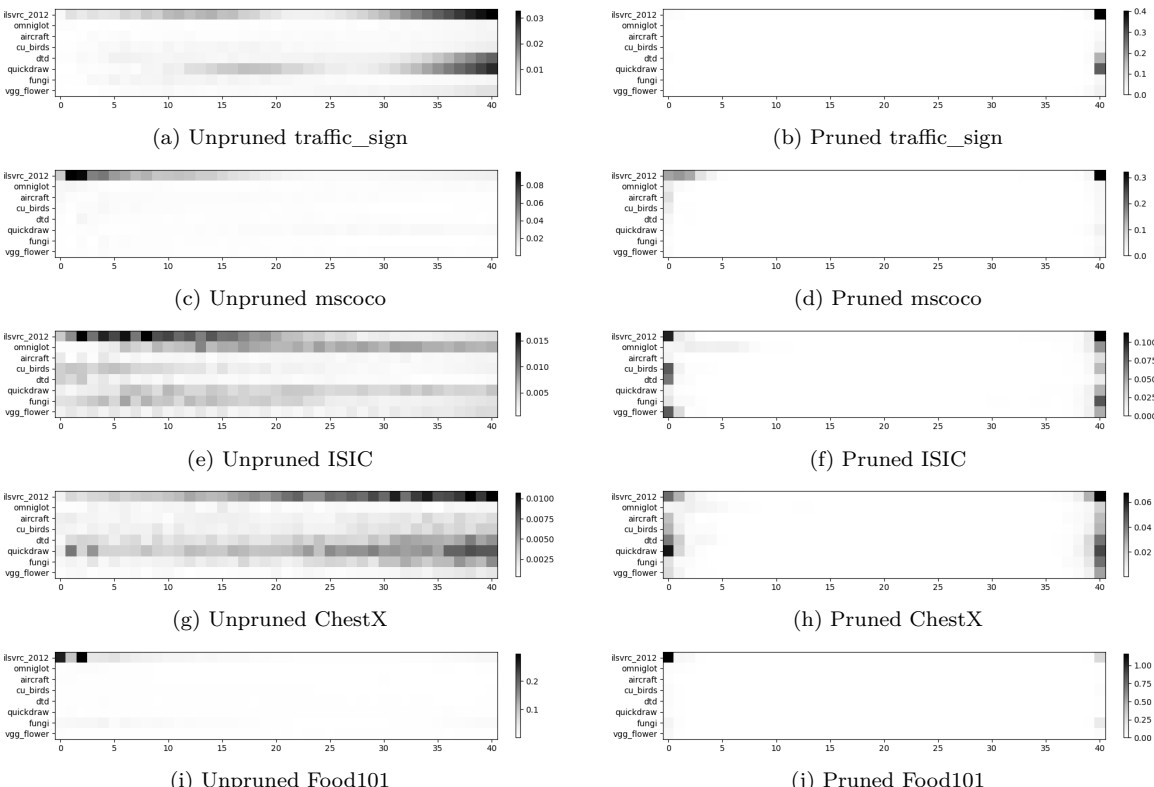

(a) Unpruned traffic_sign     (b) Pruned traffic_sign

(c) Unpruned mscoco     (d) Pruned mscoco

(e) Unpruned ISIC     (f) Pruned ISIC

(g) Unpruned ChestX     (h) Pruned ChestX

(i) Unpruned Food101     (j) Pruned Food101

Figure 2: Pairs of unpruned and pruned stacking classifier weight matrices

Table 2: Ablation study results with BSS, UFSS, and UBSS, performed with either zero or maximum patience

| Dataset | BSS P = 0 (Bi) | | BSS P = 328 (BiP) | | UFSS P = 0 (F) | | UFSS P = 328 (FP) | | UBSS P = 0 (Ba) | | UBSS P = 328 (BaP) | |
|---|---|---|---|---|---|---|---|---|---|---|---|---|
| | acc | ratio | acc | ratio | acc | ratio | acc | ratio | acc | ratio | acc | ratio |
| ilsvrc_2012 | 56.3±1.2 | 3.8±0.3 | **56.4±1.2** | 22.8±1.5 | 56.0±1.1● | 6.7±0.3 | 56.2±1.1● | 44.0±2.3 | 56.3±1.2 | 7.2±0.4 | 56.3±1.1 | 32.7±1.8 |
| omniglot | 93.1±0.7 | 12.3±0.8 | 93.3±0.7○ | 31.0±1.8 | 93.8±0.6○ | 14.3±0.8 | 93.3±0.6○ | 39.3±2.3 | 93.2±0.7 | 11.5±0.8 | 93.3±0.7○ | 32.7±2.2 |
| aircraft | **87.8±0.8** | 4.9±0.4 | 87.7±0.8 | 30.1±1.6 | **87.8±0.6** | 11.1±0.4 | 87.7±0.8 | 55.5±2.2 | 87.7±0.8 | 5.5±0.4 | 87.7±0.8 | 29.5±1.8 |
| cu_birds | 80.0±0.9 | 4.0±0.3 | **80.1±0.8** | 24.0±1.4 | 79.6±0.8● | 8.3±0.4 | 79.8±0.9● | 44.4±2.2 | 79.8±0.9● | 5.9±0.4 | 79.9±0.8 | 55.6±2.4 |
| dtd | **76.5±0.8** | 3.2±0.5 | 76.4±0.8 | 18.6±1.4 | 75.7±0.8● | 7.8±0.3 | 76.0±0.8● | 39.8±2.1 | **76.5±0.8** | 3.4±0.3 | 76.4±0.8 | 24.3±1.6 |
| quickdraw | **83.6±0.6** | 2.9±0.2 | 83.5±0.6● | 16.2±1.1 | 83.2±0.6● | 6.7±0.3 | 83.3±0.6● | 40.9±2.1 | 83.4±0.6● | 5.6±0.4 | 83.4±0.6● | 26.0±1.6 |
| fungi | 69.5±1.1 | 3.2±0.3 | 69.5±1.1 | 26.5±1.5 | **69.7±1.1○** | 7.4±0.3 | 69.5±1.1 | 42.7±2.0 | **69.7±1.1** | 6.6±0.4 | 69.6±1.1○ | 32.4±1.9 |
| vgg_flower | 91.8±0.7 | 5.6±0.5 | 91.9±0.7 | 19.9±1.4 | 91.5±0.7● | 15.1±0.6 | 91.7±0.7● | 44.7±1.9 | **92.0±0.7** | 4.9±0.5 | **92.0±0.7○** | 20.8±1.7 |
| WG avg | 79.83 | 4.99 | **79.85** | 23.64 | 79.66 | 9.68 | 79.69 | 43.91 | 79.83 | 6.33 | 79.83 | 31.75 |
| WG rank | **3.33** | | 3.35 | | 3.87 | | 3.62 | | 3.43 | | 3.39 | |
| traffic_sign | **85.7±0.9** | 2.9±0.3 | 85.4±1.0● | 30.6±1.9 | 71.9±1.1● | 4.6±0.2 | 84.5±1.0● | 85.9±1.7 | **85.7±0.9○** | 2.3±0.3 | 85.4±0.9● | 19.7±1.6 |
| mscoco | **54.5±1.0** | 3.4±0.2 | 54.3±1.0● | 43.5±2.1 | 53.0±1.0● | 3.3±0.2 | 53.8±1.0● | 76.0±2.1 | 53.7±1.0● | 4.2±0.3 | 53.9±1.0● | 69.3±2.5 |
| mnist | 97.0±0.5 | 6.4±0.6 | **97.1±0.5** | 24.0±1.4 | 97.0±0.5● | 8.7±0.5 | 96.9±0.5 | 28.4±1.7 | 97.0±0.5 | 5.5±0.6 | 97.0±0.5 | 40.5±2.1 |
| cifar10 | 78.3±0.9 | 2.7±0.2 | 78.2±0.9 | 37.8±2.2 | 74.6±0.9● | 3.1±0.2 | 77.3±0.9● | 63.2±2.6 | **78.5±0.9○** | 2.6±0.2 | 78.2±0.9 | 40.5±2.1 |
| cifar100 | **70.6±1.1** | 3.5±0.3 | 70.5±1.1● | 36.4±2.0 | 66.5±1.1● | 3.7±0.2 | 70.0±1.1● | 71.6±2.4 | **70.6±1.1** | 5.3±0.4 | 70.4±1.1● | 53.2±2.5 |
| CropDisease | 88.0±0.7 | 4.6±0.4 | **88.1±0.7○** | 34.4±1.7 | 85.4±0.7● | 5.7±0.2 | 87.7±0.7● | 60.0±2.2 | 87.8±0.7● | 4.2±0.4 | 87.9±0.7 | 40.8±1.8 |
| EuroSAT | 89.3±0.6 | 2.5±0.2 | 89.3±0.6 | 20.7±1.6 | 85.7±0.6● | 3.1±0.2 | 88.2±0.6● | 53.3±2.6 | **89.4±0.6** | 2.1±0.2 | 89.3±0.6 | 23.1±1.6 |
| ISIC | 48.3±0.9 | 3.4±0.2 | 49.0±0.9○ | 38.3±2.0 | 47.2±0.9● | 4.1±0.2 | **49.4±0.9○** | 66.1±2.2 | 47.7±1.0● | 2.5±0.1 | 48.5±0.9 | 37.1±1.8 |
| ChestX | 27.0±0.6 | 3.2±0.3 | 27.5±0.6○ | 58.5±2.2 | 25.4±0.5● | 3.6±0.2 | **27.6±0.6○** | 69.8±2.2 | 27.0±0.6 | 2.2±0.3 | 27.2±0.6 | 40.0±2.1 |
| Food101 | **55.7±1.1** | 2.9±0.2 | **55.7±1.1** | 20.1±1.7 | 53.3±1.1● | 3.7±0.1 | 54.4±1.1● | 51.7±2.8 | 54.6±1.1● | 4.6±0.3 | 55.2±1.1● | 81.2±2.1 |
| SG avg | 69.44 | 3.55 | **69.51** | 34.12 | 66.00 | 4.36 | 68.98 | 62.60 | 69.20 | 3.55 | 69.30 | 42.60 |
| SG rank | **3.03** | | 3.05 | | 4.75 | | 3.61 | | 3.32 | | 3.23 | |

Table 3: Ablation study comparing BSS to selecting the same number of random snapshots in each episode

| Dataset | BSS | random |
|---|---|---|
| ilsvrc_2012 | **56.35±1.16** | 50.87±1.35 ● |
| omniglot | **93.09±0.70** | 91.89±0.78 ● |
| aircraft | **87.75±0.79** | 80.85±1.33 ● |
| cu_birds | **80.05±0.86** | 75.26±1.10 ● |
| dtd | **76.48±0.82** | 72.13±0.94 ● |
| quickdraw | **83.56±0.59** | 77.14±1.01 ● |
| fungi | **69.50±1.14** | 60.14±1.55 ● |
| vgg_flower | **91.85±0.70** | 90.42±0.73 ● |
| WG avg | **79.83** | 74.84 |
| traffic_sign | **85.66±0.95** | 80.21±1.07 ● |
| mscoco | **54.54±1.03** | 48.54±1.18 ● |
| mnist | **96.99±0.52** | 96.54±0.51 ● |
| cifar10 | **78.32±0.89** | 68.93±1.22 ● |
| cifar100 | **70.64±1.06** | 63.57±1.29 ● |
| CropDisease | **87.96±0.68** | 86.31±0.72 ● |
| EuroSAT | **89.33±0.64** | 85.86±0.72 ● |
| ISIC | **48.32±0.94** | 47.54±0.90 ● |
| ChestX | 27.04±0.58 | **27.11±0.58** |
| Food101 | **55.71±1.12** | 48.59±1.26 ● |
| SG avg | **69.44** | 65.32 |

## 5.3 Ablation Study

Table 2 shows BSS, UFSS, and UBSS results. Each strategy is performed either with no patience or with maximum patience, which guarantees candidate pool depletion.

For both, zero and maximum patience, BSS achieves higher accuracy and snapshot reduction than the two unidirectional variants. This again indicates that snapshots relevant to a task can be at both the beginning and the end of fine-tuning and adopting the model at the end of fine-tuning may not be best in every CDFSL scenario. Given the same strategy, maximum patience leads to better accuracy but lower reduction. The mean accuracy difference between zero and maximum patience is small for BSS, while it is more substantial for UFSS and UBSS. Given these observations, we conclude that BSS without patience achieves the best balance between accuracy and reduction.

Table 3 compares BSS to a random selection strategy which matches the number of snapshots selected by BSS in each episode but samples the snapshots randomly from a uniform distribution. BSS significantly outperforms the random strategy in all domains except ChestX, where no statistically significant difference is found. This indicates that BSS outperforms the uniformly random null hypothesis in most cases. This

Table 4: Results of FES with/without STC and/or BSS

| Dataset | BSS STC acc | ratio | full sup acc | full STC acc | BSS sup acc | ratio |
|---|---|---|---|---|---|---|
| ilsvrc_2012 | **56.72**±**1.15** | 4.07±0.25 | 56.22±1.14 ● | 56.63±1.13 | 56.35±1.16 ● | 3.76±0.28 |
| omniglot | 93.68±0.67 | 13.14±0.82 | 93.33±0.64 ● | **94.00**±**0.58** ○ | 93.09±0.70 ● | 12.29±0.81 |
| aircraft | **88.01**±**0.73** | 4.88±0.32 | 87.64±0.79 ● | 87.88±0.75 | 87.75±0.79 ● | 4.86±0.36 |
| cu_birds | **80.30**±**0.82** | 4.81±0.30 | 79.93±0.85 ● | 80.17±0.82 ● | 80.05±0.86 ● | 4.03±0.26 |
| dtd | 76.46±0.77 | 3.34±0.27 | 76.21±0.81 ● | 76.18±0.77 ● | **76.48**±**0.82** | 3.23±0.26 |
| quickdraw | **83.95**±**0.57** | 3.39±0.25 | 83.38±0.61 ● | 83.85±0.58 ● | 83.56±0.59 ● | 2.93±0.18 |
| fungi | 70.86±1.10 | 3.57±0.25 | 69.43±1.13 ● | **70.88**±**1.09** | 69.50±1.14 ● | 3.24±0.26 |
| vgg_flower | 92.34±0.67 | 5.88±0.50 | 91.92±0.66 ● | **92.43**±**0.61** | 91.85±0.70 ● | 5.62±0.49 |
| WG avg | **80.29** | 5.39 | 79.74 | 80.25 | 79.83 | 4.99 |
| WG rank | **2.34** | | 2.67 | 2.35 | 2.64 | |
| traffic_sign | **86.48**±**0.95** | 2.77±0.28 | 84.90±0.97 ● | 85.99±0.96 ● | 85.66±0.95 ● | 2.85±0.31 |
| mscoco | **55.51**±**1.02** | 3.49±0.28 | 54.14±1.02 ● | 55.48±1.01 | 54.54±1.03 ● | 3.38±0.22 |
| mnist | 97.31±0.47 | 6.77±0.64 | 97.08±0.48 ● | **97.37**±**0.43** | 96.99±0.52 ● | 6.35±0.63 |
| cifar10 | **78.96**±**0.84** | 2.69±0.18 | 78.14±0.87 ● | 78.61±0.84 ● | 78.32±0.89 ● | 2.65±0.21 |
| cifar100 | **71.18**±**1.03** | 3.58±0.34 | 70.35±1.05 ● | 70.87±1.03 ● | 70.64±1.06 ● | 3.55±0.33 |
| CropDisease | 88.83±0.66 | 4.45±0.35 | 88.15±0.68 ● | **89.07**±**0.64** ○ | 87.96±0.68 ● | 4.57±0.35 |
| EuroSAT | **89.56**±**0.61** | 2.54±0.21 | 88.85±0.63 ● | 89.02±0.61 ● | 89.33±0.64 ● | 2.50±0.20 |
| ISIC | 49.66±0.95 | 3.19±0.20 | 49.52±0.93 ● | **51.43**±**0.93** ○ | 48.32±0.94 ● | 3.42±0.17 |
| ChestX | 27.16±0.61 | 3.00±0.17 | 27.67±0.61 ○ | **28.44**±**0.64** ○ | 27.04±0.58 | 3.20±0.33 |
| Food101 | **55.99**±**1.11** | 3.12±0.20 | 55.22±1.11 ● | 55.47±1.11 ● | 55.71±1.12 ● | 2.94±0.19 |
| SG avg | 70.08 | 3.56 | 69.39 | **70.18** | 69.44 | 3.55 |
| SG rank | **2.22** | | 2.83 | 2.25 | 2.71 | |

Table 5: Computational cost of the following methods, all applied with 40 iterations of TSA fine-tuning: a single URL universal feature extractor, unpruned FES using the eight Meta-Dataset source domain extractors, and the same FES setup with BSS pruning. The values are obtained using the largest traffic_sign support set (497 instances) in our evaluation.

| | single extractor | FES full | FES BSS |
|---|---|---|---|
| fine-tune time | 9.30s | 150.95s | 150.95s |
| prune time | - | - | 29.22s |
| stacking time | - | 0.33s | 0.03s |
| frozen parameters | 11M | 11M×8 = 88M | 11M×6 = 66M |
| trainable parameters | 1.5M | 1.5M×8 = 12M | 1.5M×6 = 9M |
| stored parameters | 11M + 1.5M = 12.5M | 11M×8 + 1.5M×328 = 580M | 11M×6 + 1.5M×51 = 142.5M |
| stacking classifier parameters | - | 328 | 51 |
| training GPU memory | 8.2G | 8.3G | 8.3G |

also suggests that ChestX is different from the other target domains, likely due to its lack of similarity to all source domains.

## 5.4 Semi-supervised Pruning with STC and BSS

Table 4 shows FES results with BSS pruning ($P = 0$) and STC semi-supervised learning with 1000 unlabelled instances. The results are compared to those obtained from supervised FES without pruning, as well as FES with only STC semi-supervised learning and FES with only BSS pruning. FES with BSS and STC is set as the reference method in Table 4, and the other methods are evaluated against it using the paired $t$-test. A ● indicates significantly better reference method performance and a ○ indicates significantly better performance from the other method.

Table 4 shows that semi-supervised learning with STC consistently improves FES performance in both pruned and unpruned settings. Semi-supervised FES pruned with BSS achieves slightly lower average strong generalisation accuracy than its unpruned counterpart, but with only 3.56% of the snapshots on average. Also, five of its wins are statistically significant while there are only three statistically significant losses, and its average rank is higher too.

## 5.5 Runtime and Memory Consumption

Table 5 shows computational cost and accuracy of a single URL extractor, unpruned FES, and FES with BSS pruning. The ResNet18 architecture contains 11M parameters, while TSA adaptors for one such model contain 1.5M parameters. As only the adaptors are fine-tuned, only one copy of the 11M base parameters needs to be saved for each extractor, and each snapshot only contains the 1.5M TSA adaptor parameters. Note that we use the largest traffic_sign support set, which has 497 training instances, and Table 5 represents an upper bound in our experimental setup. Fine-tuning and stacking are performed on an NVIDIA A6000 GPU, while pruning is performed on an Intel Core i7-6700K CPU as a CPU iterates through snapshot combinations more quickly than a GPU. For this particular episode, pruning leads to 51 snapshots belonging to six extractors, and the other two extractors can be excluded entirely. Note that FES can fine-tune its extractors sequentially, so its GPU memory requirement is close to using a single extractor of the same architecture.

The main time consumption of FES is due to its fine-tuning process, as the extractors need to be fine-tuned both for cross-validation and on the full support set. Time consumption of pruning is significant but not overwhelming, constituting an approximate 20% runtime increase compared to the FES version without pruning. However, the pruned version requires approximately 75% less storage space for its parameters compared to the unpruned version. BSS scales well with the number of snapshots as discussed in Section 3, while exhaustive search scales exponentially with the number of snapshots considered. Likely due to the larger number of training instances in the episode considered, the ratio of remaining snapshots, i.e., 51 out of 328, is greater than the traffic_sign average reported in Table 1. Users are advised to refer to Tables 1 and 5 to consider trade-off among accuracy, runtime, and storage cost when choosing among a universal model, unpruned FES, and FES pruned with BSS.

## 6 Conclusion

We propose BSS, a pruning strategy for FES based on linear forward selection that leverages its cross-validation process and the ordered nature of FES snapshots. BSS achieves over 95% average snapshot reduction while retaining state-of-the-art supervised and semi-supervised accuracy. It outperforms several baselines and unidirectional snapshot selection strategies, indicating that relevant snapshots can be found at both the beginning and the end of fine-tuning. We also show that visualisation of BSS models aids analysis of target domains.

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
