# OpenReview forum: "Pruning Feature Extractor Stacking for Cross-domain Few-shot Learning"
_TMLR — Accepted by TMLR_

### Review · Reviewer_RGkX · 2024-11-24

**Summary Of Contributions:**

This paper proposes a novel cross domain few shot learning method. Based on existing feature stacking approach, the author proposes to reduce the candidate feature extractors by applying bidirectional feature selection method. The proposed method demonstrates comparable accuracy to the full feature case with much lower number of feature extractor snapshot used in the inference.

**Audience:**

Yes

**Broader Impact Concerns:**

The paper has no broader impact discussion. But since this is an algorithmic paper, I think the potential negative impact of the paper is small.

**Claims And Evidence:**

No

**Requested Changes:**

More detailed discussion on the motivation, full overview of the problem setting and the novelty compared to the existing work would improve the clarity of the paper.

Discussion about the advantage of the proposed method on the last snapshot method (EL) and comparison to more baselines written in the weakness section would help to make the advantage of the proposed method more evident.

**Strengths And Weaknesses:**

Strength

The author proposes a method that applies novel bidirectional snapshot selection strategy to feature stacking based cross domain few shot learning method. Since the feature stacking method uses all the feature snapshot during training which takes much memory, the motivation to reduce the snapshots by feature selection is well understandable.

The author conducts experimental evaluation on the several standard image recognition benchmarks and compares with multiple benchmark methods. The proposed method demonstrates comparable accuracy to the baseline that uses full snapshots with much less computational memory.

The author conducts several ablation study, visualizations and measurement of the computation cost to investigate the effectiveness of the proposed method.


Weakness

The proposed method depends much on the existing methods (e.g. Wang et al., 2024) and the novelty is small.
Further, since the paper only explains the proposed bidirectional snapshot selection strategy in detail, it is difficult to understand the whole few shot learning pipeline without reading such previous papers.
It would be better to make the paper self-contained by adding the explanation about the problem setting of the cross domain few shot learning in pretraining, fine-tuning, and inference phase and about the feature extractor stacking approach in more detail.

The motivation why adopting bidirectional strategy is unclear. It would be more discussed in the introduction and the proposed method section.

As for the experimental comparison, it seems last snapshot method (EL) demonstrates not so bad accuracy with lower cost than the proposed bidirectional snapshot selection strategy. Is there clear benefit of the proposed method compared to EL?

Since the motivation is the reduce in the inference cost, I would like to see the comparison with the full snapshot method with some sparse method in the final combination weight training stage (e.g. l1 regularization). Further, from Figure 2, most of the weights are assigned to first and last snapshots. Therefore, I would also like to see the comparison to the exhaustive search through first and last snapshot as an extension of EL.

---

> ### Author Response · Authors · 2025-01-17
> **Rebuttal**
>
> Our response to the weaknesses and how we have revised the manuscript accordingly are as follows:
>
> > The proposed method depends much on the existing methods (e.g. Wang et al., 2024) and the novelty is small. Further, since the paper only explains the proposed bidirectional snapshot selection strategy in detail, it is difficult to understand the whole few shot learning pipeline without reading such previous papers. It would be better to make the paper self-contained by adding the explanation about the problem setting of the cross domain few shot learning in pretraining, fine-tuning, and inference phase and about the feature extractor stacking approach in more detail.
>
> We appreciate the concerns regarding the limited novelty of the method we propose, as what we present is clearly an incremental improvement on an existing method. However, according to the official acceptance criteria of TMLR at https://jmlr.org/tmlr/acceptance-criteria.html, the key question is whether *some individuals in TMLR's audience* would be *interested* in the findings of this paper. Importantly, a higher threshold on novelty, significance, and impact is explicitly not applicable to this journal according to the acceptance criteria. Our submission presents an incremental improvement of an existing method that we believe readers of the original work and machine learning practitioners who would like to apply the method would find interesting.
>
> We have added a paragraph after the first paragraph in Section 2.1 to explain the three stages of CDFSL. We have also added an overview of FES to the beginning of the first paragraph of Section 2.2 for clarity. However, the FES paper should be referred to for a comprehensive explanation of the FES algorithm.
>
> > The motivation why adopting bidirectional strategy is unclear. It would be more discussed in the introduction and the proposed method section.
>
> Section 1 states storage cost reduction as the main motivation for BSS. We have added a paragraph to the end of Section 3 to analyse the complexity of BSS in relation to the number of snapshots and discuss its advantage in complexity over naive greedy selection.
>
> > As for the experimental comparison, it seems last snapshot method (EL) demonstrates not so bad accuracy with lower cost than the proposed bidirectional snapshot selection strategy. Is there clear benefit of the proposed method compared to EL?
>
> As stated in the second paragraph of Section 5.1, BSS achieves higher average strong generalisation accuracy than unpruned FES while the two exhaustive baselines perform worse, although as noted by the reviewer, the differences may not be substantial. We have added a paragraph to the end of Section 3 and revised the second paragraph of Section 5.5 to discuss the complexity and scalability advantages of BSS in relation to the number of snapshots compared to exhaustive search and naive greedy selection.
>
> > Since the motivation is the reduce in the inference cost, I would like to see the comparison with the full snapshot method with some sparse method in the final combination weight training stage (e.g. l1 regularization). Further, from Figure 2, most of the weights are assigned to first and last snapshots. Therefore, I would also like to see the comparison to the exhaustive search through first and last snapshot as an extension of EL.
>
> We experimented with L1 regularisation for pruning early on, but the results were discouraging in both the accuracy and the snapshot reduction achieved, which is why we have not further pursued regularisation as a means to pruning. We have added this to the list of baselines we have tried without obtaining remarkable results reported in the last paragraph of Section 5.1. We acknowledge that exhaustively searching through the first and last snapshots may return competitive accuracy, but the cost of the experiment is prohibitive because exhaustive search scales exponentially with the number of snapshots considered. Note that we did evaluate exhaustive search involving only each extractor's snapshot with the lowest individual cross-validation loss, as mentioned in the third paragraph of Section 5.1. We have added a paragraph to the end of Section 3 to discuss the complexity of BSS. We have also revised the second paragraph of Section 5.5 to reflect that BSS scales more favourably to the number of snapshots compared to exhaustive search.

---

### Review · Reviewer_nLF2 · 2024-12-20

**Summary Of Contributions:**

The problem addressed in this paper is cross-domain few-shot learning (CDFSL). The work explores the use of a weighted averaging classifier derived from multiple source domain-specific feature extractors. In this category of methods, many existing approaches rely on a straightforward weighted averaging across numerous feature extractors, which can be computationally expensive. To address this overhead, the paper proposes a pruning approach that selects a subset of feature extractors while maintaining classification performance.

The proposed method is relatively simple: it iteratively selects candidates from multiple feature extractors based on a classification metric obtained through cross-validation. Despite its simplicity, this approach is effective. Experiments on the Meta-Dataset demonstrate that the proposed pruning strategy achieves comparable generalization performance while significantly reducing the number of parameters and improving inference throughput.

**Audience:**

No

**Claims And Evidence:**

No

**Requested Changes:**

I suggest that the manuscript should be revised. In Section 3.2, the first sentence states that Figure 1 and Algorithm 1 present the main idea, but it does not provide sufficient detail, making it difficult to understand the overall method. While the following sentences do describe the method in more detail, I personally recommend revising the first paragraph to clearly present the main idea and emphasize an overview of the method. This improvement would enhance the clarity and reliability of the manuscript.

Although pruning 328 snapshots into a smaller subset is a promising approach for significantly enhancing inference throughput, its broader importance and applicability could be better demonstrated through experiments on a larger number of snapshots or by utilizing more complex feature extractors. Additionally, replacing ResNet-18 with a more robust and powerful feature extractor could more effectively highlight the advantages of the proposed pruning method in terms of memory and throughputs.

**Strengths And Weaknesses:**

**Strengths**
Pruning unnecessary information from multiple extractors for meta-classifiers is a reasonable approach. The proposed method is both simple and straightforward, effectively determining which extractors should be selected to maintain performance.

**Weaknesses**
According to the TMLR guidelines, a key criterion for evaluation is whether the problem addressed holds substantial importance for the broader research community. In this case, the paper focuses on reducing storage and inference costs in a CDFSL setting, but the significance of this contribution is not entirely clear. Furthermore, the small-scale experiments presented are not sufficiently convincing to establish that pruning snapshots genuinely offers meaningful computational benefits.

Regarding the methodology, why does the search start with only the first and last snapshots instead of considering the entire set of snapshots? While a bidirectional search certainly reduces the computational effort during pruning, examining all available snapshots could potentially allow for more aggressive pruning without harming overall performance. Naturally, this broader approach would increase the complexity of the pruning process during training, as we would need to evaluate which candidate pruning step leads to the greatest reduction in cross-validation loss. However, in my view, the primary advantage of pruning lies in improving inference throughput, even if this comes at the expense of a more intensive training process.

---

> ### Author Response · Authors · 2025-01-17
> **Rebuttal**
>
> Our response to the weaknesses are as follows:
>
> > According to the TMLR guidelines, a key criterion for evaluation is whether the problem addressed holds substantial importance for the broader research community. In this case, the paper focuses on reducing storage and inference costs in a CDFSL setting, but the significance of this contribution is not entirely clear. Furthermore, the small-scale experiments presented are not sufficiently convincing to establish that pruning snapshots genuinely offers meaningful computational benefits.
>
> We respectfully disagree with the statement that “According to the TMLR guidelines, a key criterion for evaluation is whether the problem addressed holds substantial importance for the broader research community.”  We invite the reviewer to revisit the official acceptance criteria of TMLR at https://jmlr.org/tmlr/acceptance-criteria.html. The key question stated on that page is whether *some individuals in TMLR's audience* would be *interested* in the findings of this paper. Our submission presents an incremental improvement of an existing method that we believe readers of the original work and machine learning practitioners who would like to apply the method would find interesting.
>
> The experimental framework used in our submission is consistent with existing CDFSL literature. Computational cost is reported and discussed in Section 5.5.
>
> > Regarding the methodology, why does the search start with only the first and last snapshots instead of considering the entire set of snapshots? While a bidirectional search certainly reduces the computational effort during pruning, examining all available snapshots could potentially allow for more aggressive pruning without harming overall performance. Naturally, this broader approach would increase the complexity of the pruning process during training, as we would need to evaluate which candidate pruning step leads to the greatest reduction in cross-validation loss. However, in my view, the primary advantage of pruning lies in improving inference throughput, even if this comes at the expense of a more intensive training process.
>
> We have added a paragraph to the end of Section 3 analysing the computational complexity of BSS and comparing it to naive greedy selection from all snapshots. The reviewer's point regarding inference throughput is a valid one, but we leave potential throughput-oriented methods for future work. We present BSS in this submission with its main advantage of significant storage reduction. However, it is worth noting that by reducing the snapshot count of FES to 5% of the original, BSS theoretically attains approximately 20 times the inference throughput of unpruned FES.
>
> ---
>
> Our response to the requested changes are as follows:
>
> > I suggest that the manuscript should be revised. In Section 3.2, the first sentence states that Figure 1 and Algorithm 1 present the main idea, but it does not provide sufficient detail, making it difficult to understand the overall method. While the following sentences do describe the method in more detail, I personally recommend revising the first paragraph to clearly present the main idea and emphasize an overview of the method. This improvement would enhance the clarity and reliability of the manuscript.
>
> We have revised the first paragraph of Section 3 to provide an overview of BSS.
>
> > Although pruning 328 snapshots into a smaller subset is a promising approach for significantly enhancing inference throughput, its broader importance and applicability could be better demonstrated through experiments on a larger number of snapshots or by utilizing more complex feature extractors. Additionally, replacing ResNet-18 with a more robust and powerful feature extractor could more effectively highlight the advantages of the proposed pruning method in terms of memory and throughputs.
>
> We acknowledge the prospect of applying BSS to different feature extractors and snapshot sets, but we believe it is beyond the scope of this work. The experimental setup in this submission is consistent with prior literature including the FES paper to provide a controlled comparison between BSS-pruned FES and unpruned FES. As mentioned before, we have added a paragraph to the end of Section 3 to discuss the complexity advantage of BSS in relation to snapshot sequence length.

---

### Review · Reviewer_t8j9 · 2025-01-14

**Summary Of Contributions:**

This manuscript proposes an effective pruning strategy, Bidirectional Snapshot Selection (BSS), for Feature Extractor Stacking (FES) in Cross-domain Few-shot Learning (CDFSL). FES has demonstrated state-of-the-art performance by stacking multiple model snapshots from source domain extractors. However, FES suffers from the significant storage costs due to the retention of snapshots after each fine-tuning iteration. The authors address this issue by leveraging the ordered nature of the snapshots and a bidirectional pruning method, which can reduces the snapshot storage by over 95% while maintaining the same level of accuracy.

**Audience:**

Yes

**Claims And Evidence:**

Yes

**Requested Changes:**

1. Based on the similarities between FES and active learning in selecting the most informative examples, I suggest expanding the literature review to include a more detailed discussion of active learning. Specifically, including methods such as uncertainty sampling or query-by-committee could help situate FES within the broader landscape of model selection and optimization strategies.

2. The manuscript would benefit from a deeper theoretical explanation of why the bidirectional pruning strategy is effective. A more formal analysis of the underlying principles would help strengthen the contribution and allow readers to better understand the mechanism behind its success.

3. The manuscript could explore and discuss the performance of BSS in scenarios where source and target domains are weakly related (such as ISIC and ChestX). A more thorough examination of potential limitations in these edge cases would provide a more comprehensive view of the method’s applicability across various domains.

4. My main concern is with the overall contribution and novelty of the study. While FES demonstrates strong performance, its high computational overhead compared to other CDFSL methods is a significant drawback, particularly when the performance improvements are not significantly larger. The study appears to have limited applicability, as it is entirely dependent on FES. I suggest that the motivations for focusing on FES be clearly stated. Merely claiming it achieves state-of-the-art performance is not sufficient justification for the study. A more detailed explanation of why FES is chosen and its practical relevance would strengthen the contribution.

**Strengths And Weaknesses:**

### Strengths:

1. The introduction of Bidirectional Snapshot Selection (BSS) for pruning Feature Extractor Stacking (FES) is an effective solution to reduce storage requirements while preserving model accuracy.

2. BSS reduces storage and computational costs of FES, which makes it suitable for real-world applications with limited resources.

3. Comprehensive Evaluation: The method is rigorously evaluated on the Meta-Dataset benchmark across various target domains, highlighting its general applicability.

4. The use of heatmap visualizations to display the weights assigned to snapshots offers clear insights into the model’s behaviors, which enhances interpretability.

### Weaknesses:

1. Based on my understanding, FES, by retaining and combining model snapshots based on their performance through cross-validation (query and support sets), is very similar to the settings of active learning in selecting the most informative examples. A more detailed discussion of active learning, including methods like uncertainty sampling or query-by-committee, could improve the paper’s literature review by situating FES within the broader landscape of model selection and optimization strategies. This would provide readers with a clearer understanding of how FES relates to and differs from active learning approaches.

2. The manuscript lacks a detailed theoretical explanation of why the bidirectional pruning strategy works effectively, which could provide a deeper understanding of its underlying principles.

3. Limited Discussion on Edge Cases: The performance of BSS in domains where source and target domains are weakly related (e.g., ISIC and ChestX) is not thoroughly explored, and the paper does not address potential limitations in such scenarios.

4. My main concern is the contribution and novelty of the study. While FES demonstrates strong performance, its significant computational overhead can be a major drawback compared to other CDFSL methods, especially when the performance lifts are not substantially greater. Given the high storage costs and computational complexity, further studies focused on FES may face challenges in terms of efficiency and practicality.

---

> ### Author Response · Authors · 2025-01-17
> **Rebuttal**
>
> Our response to the weaknesses and how we have revised the manuscript accordingly are as follows:
>
> > Based on my understanding, FES, by retaining and combining model snapshots based on their performance through cross-validation (query and support sets), is very similar to the settings of active learning in selecting the most informative examples. A more detailed discussion of active learning, including methods like uncertainty sampling or query-by-committee, could improve the paper’s literature review by situating FES within the broader landscape of model selection and optimization strategies. This would provide readers with a clearer understanding of how FES relates to and differs from active learning approaches.
>
> We appreciate the suggestion but believe covering active learning in detail is beyond the scope of our paper. However, we have added references to semi-supervised learning and active learning in the introduction to the section covering related work.
>
> > The manuscript lacks a detailed theoretical explanation of why the bidirectional pruning strategy works effectively, which could provide a deeper understanding of its underlying principles.
>
> We have added a paragraph to the end of Section 3 analysing the computational complexity of BSS. For theoretical explanations of the effectiveness of wrapper-based feature selection, please refer to the original publications referenced in our manuscript.
>
> > Limited Discussion on Edge Cases: The performance of BSS in domains where source and target domains are weakly related (e.g., ISIC and ChestX) is not thoroughly explored, and the paper does not address potential limitations in such scenarios.
>
> We have revised the penultimate paragraph in Section 5.2 to discuss the potential connection between domain relations and BSS performance in more detail.
>
> > My main concern is the contribution and novelty of the study. While FES demonstrates strong performance, its significant computational overhead can be a major drawback compared to other CDFSL methods, especially when the performance lifts are not substantially greater. Given the high storage costs and computational complexity, further studies focused on FES may face challenges in terms of efficiency and practicality.
>
> As the reviewer’s “main concern is the contribution and novelty of the study”, we would like to refer to the official acceptance criteria of TMLR at https://jmlr.org/tmlr/acceptance-criteria.html. The key question is whether *some individuals in TMLR's audience* would be *interested* in the findings of this paper. Importantly, a higher threshold on novelty, significance, and impact is explicitly not applicable to this journal according to the acceptance criteria. Our submission presents an incremental improvement of an existing method that we believe readers of the original work and machine learning practitioners who would like to apply the method would find interesting.

---

### Decision · Action_Editor_dPrX · 2025-04-11

**Recommendation:** Accept with minor revision

**Comment:**

The core contribution of this paper is a bidirectional search method to prune snapshots from the feature extractor stacking (FES) method of Wang et al for cross-domain few-shot learning (CDFL). It's a pretty simple approach that seems quite effective at reducing the memory and inference footprint of FES.

The reviewers were mainly concerned about the novelty and significance aspect of the work. It is a rather specific approach to a fairly esoteric method. For those interested in CDFL and FES specifically, this could help with practicality. There may not be much interest from a more general audience. More work on general insight and a larger scale could help with this. Otherwise, if we restrict ourselves to the narrow scope of improving storage costs for FES, then the claims are justified.

I would like to see a comparison against a random baseline, where a random subset of snapshots are selected equal to the size of the subset found by BSS.

**Audience:**

I think this will be of interest to a fairly small segment of researchers doing CDFSL with an interest in FES, but there will be some that find this interesting.

**Claims And Evidence:**

The claims and evidence are ok. It reduces the storage cost of FES while maintaining accuracy (except in one case, which is discussed in the paper).